# Protein–ligand binding with the coarse-grained Martini model

Paulo C. T. Souza [1,4✉], Sebastian Thallmair [1,4], Paolo Conflitti [2], Carlos Ramírez-Palacios [1], Riccardo Alessandri [1], Stefano Raniolo [2], Vittorio Limongelli [2,3✉] & Siewert J. Marrink [1✉]

The detailed understanding of the binding of small molecules to proteins is the key for the development of novel drugs or to increase the acceptance of substrates by enzymes. Nowadays, computer-aided design of protein–ligand binding is an important tool to accomplish this task. Current approaches typically rely on high-throughput docking essays or computationally expensive atomistic molecular dynamics simulations. Here, we present an approach to use the recently re-parametrized coarse-grained Martini model to perform unbiased millisecond sampling of protein–ligand interactions of small drug-like molecules. Remarkably, we achieve high accuracy without the need of any a priori knowledge of binding pockets or pathways. Our approach is applied to a range of systems from the well-characterized T4 lysozyme over members of the GPCR family and nuclear receptors to a variety of enzymes. The presented results open the way to high-throughput screening of ligand libraries or protein mutations using the coarse-grained Martini model.

[1] Groningen Biomolecular Sciences and Biotechnology Institute and Zernike Institute for Advanced Materials, University of Groningen, Nijenborgh 7, 9747 AG Groningen, Netherlands. [2] Faculty of Biomedical Sciences, Institute of Computational Science, Università della Svizzera italiana (USI), via G. Buffi 13, CH-6900 Lugano, Switzerland. [3] Department of Pharmacy, University of Naples "Federico II", via D. Montesano 49, I-80131 Naples, Italy. [4] These authors contributed equally: Paulo C. T. Souza, Sebastian Thallmair. ✉email: paulocts@gmail.com; vittoriolimongelli@gmail.com; s.j.marrink@rug.nl

Protein–ligand binding interaction is fundamental to a large variety of cellular functions, including enzymatic reactions, catalysis, signal transduction, and regulation. The importance of protein–ligand interactions explains the ongoing interest to redesign these interactions, conferring novel functions or finding suitable drugs and molecular targets[1]. A typical binding pocket consisting of 10 residues gives rise to $20^{10}$ possible mutations; together with an estimated amount of $10^{60}$ potential drug-like compounds, this composes a vast chemical space. Therefore, the potential for rational design is enormous. To harness this potential, a lot of progress has been made in the development of high-throughput experimental screening techniques together with computational methods[2–5]. In this process, a significant increase of the drug design success rate arises out of the development of novel computational strategies.

Current approaches typically rely on docking assays to either predict or optimize the ligand-binding mode[6–8]. Although docking-based methods allow for high-throughput screening of large compound and/or protein mutant libraries, the accuracy of the predictions is limited. The main sources of limitation are the use of simplified energy ("scoring") functions as well as the limited treatment of protein and ligand flexibility and solvation models. In the case of protein engineering studies, only local mutations around the binding/catalytic site can benefit of docking-based methods. To alleviate these shortcomings, molecular dynamics (MD) simulation has become a popular tool in the field of drug design and discovery[9,10]. The MD technique is based on detailed interaction potentials (force fields) and, in principle, includes relevant dynamics of protein, ligand, and solvent provided that enough sampling can be achieved. In some cases, a few direct binding events of ligands can be simulated using brute force MD[11–13], but typically sampling of the relevant degrees of freedom is a limiting factor. A variety of enhanced sampling methods, such as replica exchange, funnel-metadynamics, or adaptive sampling, exist to improve the sampling and to study the binding kinetics and pathways[14–17]. In addition, when the binding mode is known, calculations can be performed to obtain (relative) binding free energies[18–21]. In practice, nowadays docking and MD are often combined to increase the accuracy of the former method[4,10,22,23]. Fully atomistic MD simulations, however, are computationally too expensive to allow for high-throughput applications.

A potential solution is the use of coarse-grained (CG) force fields, which reduce the computational cost by uniting groups of atoms into effective interaction sites resulting in a substantial computational speed-up[24,25]. The Martini model[26] is among the most popular CG force fields, and has been applied to study a wide range of biomolecular processes including successful prediction of protein–lipid binding modes[27]. In some cases, binding of lipids to sites deeply buried inside the proteins were recovered by brute force MD[28,29]. Examples of protein–ligand binding with Martini are, however, still scarce[30–32]. The question remains, therefore, whether such CG models can be applied to capture protein–ligand binding in general, including small organic compounds, such as enzyme substrates, receptor ligands, drugs, or pesticides.

Here we show that the recently re-parameterized Martini model[33] can perform this task with high accuracy, based on unbiased sampling. For all the simulations presented here, the ligands (depicted together with the CG mapping in Supplementary Fig. 1) were initially positioned randomly in the solvent. We use three different classes of examples to illustrate the potential power of this approach. First, we show millisecond long sampling of the reversible binding of seven different ligands (binders as well as non-binders) to mutants of T4 lysozyme, a well-studied protein for which ample experimental and all-atom MD simulation data

are available for comparison. Second, we show spontaneous binding and unbinding of both agonist and antagonist ligands to the adenosine $A_{2A}$ receptor ($A_{2A}R$) and adrenergic $\beta_2$ receptor ($\beta 2AR$), two different members of the membrane-embedded protein family of G protein-coupled receptors (GPCRs), as well as to the nuclear receptor farnesoid X receptor (FXR), all representing prominent pharmacological targets. Third, we demonstrate the versatility of our approach by simulating ligand binding to different enzymes, namely binding of a substrate to a complex catalytic site of an aminotransferase, which is an important representative of biocatalytic applications and binding of drugs to two members of the kinase family, proto-oncogene tyrosine-protein kinase (c-Src) and the AP2-associated protein kinase 1 (AAK1) involved in virus endocytosis and a potential target for COVID-19 treatment[34].

## Results

**Binding of ligands to T4 lysozyme**. T4 lysozyme serves as a benchmark system to investigate ligand binding[18]. Notably, the L99A mutant is known to bind hydrophobic ligands into a well-defined cavity in the core of the C-terminal domain[35]. To investigate the ligand binding to the L99A mutant of T4 lysozyme, the protein is embedded in a cubic box with 10 nm edge length containing ~8850 CG water beads (corresponding to 35,400 water molecules) and one ligand molecule, as shown in Fig. 1a. Thus, the ligand concentration is around 1.6 mM. Figure 1b–e summarizes the results obtained for the seven examined ligands benzene, phenol, indole, thieno-pyridine, toluene, ethylbenzene, and n-propylbenzene with a sampling time of 0.9 ms each (30 independent trajectories of 30 μs per ligand). In addition, the binding of benzene and toluene to the L99A/M102Q double mutant of T4 lysozyme was studied. The additional M102Q mutation reduces the hydrophobic character of the binding pocket[36] which is expected to reduce the affinity to benzene and toluene.

In case of benzene and the L99A T4 lysozyme, each individual simulation of 30 μs shows between 2–9 binding and unbinding events. In total, 156 binding and 147 unbinding events are observed (Table 1). The crystal structure of L99A T4 lysozyme including a bound benzene molecule (green) is depicted in Fig. 1b. In addition, the figure contains several CG snapshots of benzene taken from our binding simulations (red) and the benzene density in the binding pocket (transparent red isosurface). A comparison of the benzene position in the crystal structure with the CG snapshots as well as the averaged benzene density clearly shows that the CG force field Martini properly locates the buried hydrophobic binding pocket. The experimentally observed binding pose of the ligand in the binding pocket is excellently reproduced with a root mean square deviation (RMSD) of 1.4 ± 0.2 Å, which is similar to the one from atomistic studies[37]. Fig. 1c visualizes the benzene density around L99A T4 lysozyme. The blue, cyan, red, and violet isosurfaces represent occupancies which are 10, 100, 1000, and 10,000 fold higher than in water. A binding channel indicated by the blue isosurface leads from the surface of the protein toward the buried binding pocket. The density inside the binding pocket (red isosurface) is about 1000 fold higher compared to the bulk.

Figure 1d unravels the existence of more than just one binding/unbinding path which could be already easily spotted by eye based on the overall ligand densities shown in Fig. 1c. The benzene density in the vicinity of the binding pocket as well as snapshots of the benzene molecules show four binding/unbinding paths in total. Note that compared to Fig. 1b, c the protein is rotated to enable the distinction of the different paths. The high density channel from Fig. 1c corresponds to path 1 which also has

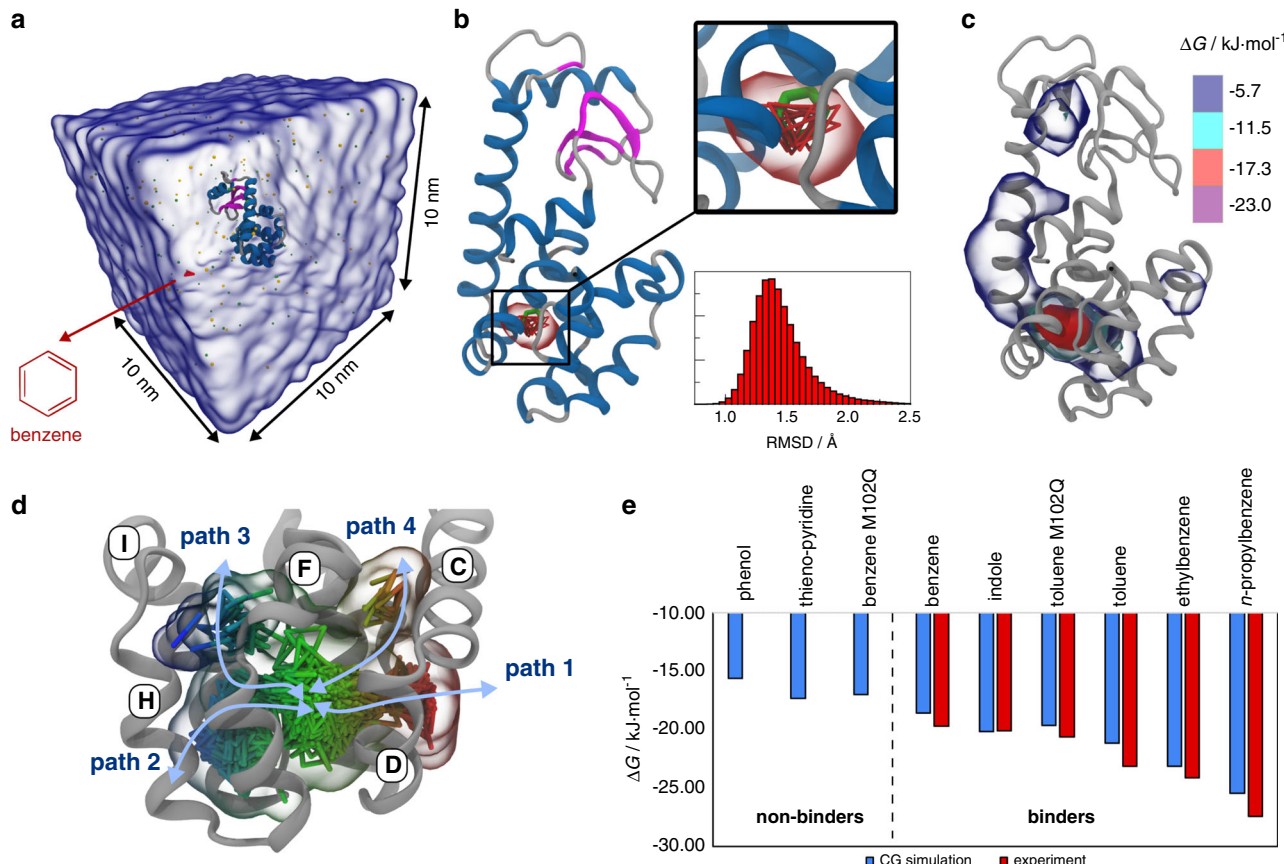

**Fig. 1 Unbiased simulations of ligand binding to T4 lysozyme at the CG Martini level. a** Simulation box containing the L99A mutant of T4 lysozyme and one benzene molecule (red) solvated in water (transparent blue surface). **b** Crystal structure of the L99A T4 lysozyme with benzene (green) in its binding pocket (pdb code: 181L[43]). In addition, several CG snapshots of benzene (red) and the benzene density in the binding pocket (transparent red isosurface) are shown. The histogram of the RMSD of benzene and the contact protein beads is depicted on the lower right. **c** Benzene density around L99A T4 lysozyme obtained from averaging 0.9 ms of CG simulations. The blue, cyan, red, and violet isosurfaces correspond to a 10, 100, 1,000, and 10,000 fold higher benzene density than in water. These densities translate to the free energy values shown at the color map. The experimental binding free energy of benzene is between −21.7 kJ/mol[98] and −17.7 kJ/mol[99]. **d** The benzene density and corresponding snapshots in the vicinity of the binding pocket reveal four binding/unbinding pathways. **e** Binding free energies calculated from the 0.9 ms of unbiased simulations for all ligands simulated here (blue) in comparison with experimental data (red). The re-parametrized Martini force field can separate non-binders from binders (dashed line). The error bars of the simulated data are <0.7 kJ/mol and not depicted here.

**Table 1 Number of binding/unbinding events, binding free energies and binding poses.**

| Ligands[a] | Non-binders | | | Binders | | | | | |
|---|---|---|---|---|---|---|---|---|---|
| | **Phenol** | **Thieno-pyridine** | **Benzene M102Q** | **Benzene** | **Indole** | **Toluene M102Q** | **Toluene** | **Ethyl-benzene** | **n-propyl-benzene** |
| Min./max. binding events per 30 μs | 0/4 | 0/3 | 2/15 | 2/9 | 0/5 | 2/10 | 1/8 | 1/5 | 1/5 |
| Min./max. unbinding events per 30 μs | 0/4 | 0/3 | 2/15 | 2/9 | 0/5 | 1/10 | 0/8 | 0/5 | 0/4 |
| Total number of binding events | 37 | 41 | 245 | 156 | 59 | 148 | 83 | 67 | 68 |
| Total number of unbinding events | 37 | 29 | 238 | 147 | 43 | 132 | 60 | 44 | 43 |
| $\Delta G_{bind}^{CG}$ [kJ mol$^{-1}$][b] | −15.6 | −17.3 | −17.0 | −18.6 | −20.2 | −19.6 | −21.2 | −23.1 | −25.4 |
| $\Delta G_{bind}^{exp}$ [kJ mol$^{-1}$][c] | — | — | — | −21.7/−17.7 | −20.5/−19.7 | −20.6 | −23.1 | −24.1 | −27.4 |
| RMSD [Å][d] | 1.6 ± 0.2 | 2.1 ± 0.2 | 1.4 ± 0.2 | 1.4 ± 0.2 | 2.0 ± 0.2 | 1.7 ± 0.3 | 1.7 ± 0.2 | 1.9 ± 0.3 | 1.9 ± 0.4 |
| Reference pdb code[e] | 1LI2 | 185 L | 181 L | 181 L | 185 L | 4I7K | 4W53 | 4W54 | 4W55 |

[a]Ligand names containing "M102Q" indicate the systems simulated with the L99A/M102Q double mutant of T4 lysozyme. The rest of the MD simulations were performed with the single mutant L99A.
[b]Binding free energies ($\Delta G_{bind}^{CG}$) are computed from radial ligand-receptor potentials of mean force obtained from unbiased MD simulations, as described in Methods.
[c]The first experimental binding free energy ($\Delta G_{bind}^{exp}$) corresponds to calorimetric data taken from refs. [18,44,97,98]. Where a second value is given, this is taken from NMR experiments[99].
[d]The RMSD is calculated for the binding pocket residues and ligand after aligning the binding pocket residues to the respective crystal structure (for details see Supplementary Methods).
[e]Crystal structures taken from ref. [43]. (benzene, indole) and [45] (toluene, ethylbenzene, and n-propylbenzene). Because no crystal structures are available for non-binders, the experimental binding mode of phenol M102Q[44], indole[43], and benzene[43] were used as references the structure to compute the RMSD for phenol, thieno-pyridine, and benzene M102Q, respectively.

been found to be the dominant dissociation path in a weighted ensemble approach[38] and the second dominant dissociation pathway in a biased MD study[39], both at atomistic resolution. The other three paths have been reported previously as well in biased[38–41] and unbiased[37,42] atomistic MD studies. An example of a binding event is provided in Supplementary Movie 1.

The ligand densities for the other ligands as well as the L99A/M102Q double mutant are depicted in Supplementary Fig. 2. Also in these eight cases, the hydrophobic binding pocket shows the highest occupancy, in agreement with the crystal structures of T4 lysozyme with the respective ligands[43–45]. The binding channel density representing path 1 is very similar to the one of benzene for all ligands except phenol, for which this region is less populated. The experimentally observed binding poses are well reproduced with an average RMSD ≤ 2.1 Å for each of the eight examples (see Table 1, Supplementary Fig. 3, and Supplementary Table 1). In the case of thieno-pyridine, indole, toluene, ethylbenzene, and n-propylbenzene, the RMSD distributions show two binding modes (Supplementary Fig. 3).

Figure 1e compares the binding free energies obtained from our simulations $\Delta G_{bind}^{CG}$ with the available experimental binding free energies $\Delta G_{bind}^{exp}$ (see also Table 1). The $\Delta G_{bind}^{CG}$ values are calculated by integrating the one-dimensional potentials of mean force depicted in Supplementary Fig. 4. Based on the simulated results, it is possible to distinguish between non-binding (three) and binding protein-ligand combinations (six). Moreover, $\Delta G_{bind}^{CG}$ is in very good agreement with the experimental values with a mean absolute error of 1 kJ/mol and a maximum error of 2 kJ/mol. To the best of our knowledge, no other study is available to date with such a good agreement of theoretical and experimental binding free energies for this range of T4 lysozyme ligands.

Our results for T4 lysozyme demonstrate the capability to accurately predict the ligand-binding pocket as well as the diverse binding pathways with the recently parametrized Martini 3 force field. Remarkably, as the results were obtained using unbiased simulations, no a priori knowledge of the binding pocket is required. Moreover, a nearly quantitative agreement of the binding free energy for all nine examined systems including different ligands and protein mutants with experimental data is achieved.

**Binding to membrane and nuclear receptors**. As a second showcase, we present the binding of two small organic molecules to a pharmacologically relevant membrane protein, namely the adenosine $A_{2A}$ receptor ($A_{2A}R$). It belongs to the class A of the GPCRs. The GPCR superfamily is a major therapeutic target whose functioning regulates several physiological processes such as vision, smell, taste, cardiovascular, neurological, and reproductive mechanisms, and $A_{2A}R$ in particular has been recognized as a drug target for the treatment of many diseases, including cancer, Parkinson's and Alzheimer's disease[46,47]. Here, we examine the ability of the Martini 3 force field to identify the crystallographic binding poses of the endogenous agonist adenosine and the natural antagonist caffeine. The recognition of the $A_{2A}R$ binding site by the ligands is challenging, since the access to the binding pocket is narrow and regulated by the extracellular loops like extracellular loop 2 (ECL2). In $A_{2A}R$, ECL2 contains a small helix that is partially folded over the ligand entry site[48]. Furthermore, interaction of the ligands with either ECL1, ECL2, or ECL3 is relevant to the binding kinetics as proved in different experimental and computational studies[13,49]. In addition to the molecular details of the pocket entrance, the protein environment is more complex, because $A_{2A}R$ is a transmembrane protein. Thus, the ligand can also be sorted to the lipid bilayer.

For each system ($A_{2A}R$—adenosine and $A_{2A}R$—caffeine) we performed 12 simulations with variable number of ligands, spanning from 7 to 13, in a cubic box with 12 nm edge length (Fig. 2a). The binding poses of adenosine and caffeine, after back-mapping to all-atom resolution[50], are compared with the ones in the crystal structures 2YDO[51] and 3RFM[52], respectively. Figure 2b shows all binding poses of adenosine with an RMSD < 2.6 Å. The RMSD distribution of all observed binding poses is depicted in Fig. 2c (top) and has an average of 3.3 ± 0.5 Å. The best binding pose obtained from the simulations has an RMSD of 2.2 Å (Fig. 2d). Comparing this pose to the crystal structure[51], we note that adenosine is shifted toward the transmembrane helix 7 (TM7), with a slight tilt of 14°. Notably, the best binding mode closely resembles the experimental one, engaging all the interactions observed in the crystallographic structure (for details see Supplementary Discussion)[51]. In the case of the $A_{2A}R$—caffeine system, we observe 142 binding events, which lead to a well-characterized binding mode. Figure 2c (bottom) shows the RMSD distribution of all binding poses, which has an average of 3.4 ± 0.5 Å. Figure 2e depicts the best binding mode which has a RMSD of 1.9 Å and is remarkably similar to the crystallographic one, being shifted by only 1.4 Å toward TM3 and tilted by 4° (for details see Supplementary Discussion)[52]. The larger number of binding events observed for caffeine with respect to adenosine can be attributed to the planar geometry of the former that better inserts into the gorge forming the entry of the $A_{2A}R$ binding site. Moreover, the two ligands show a different way to approach to $A_{2A}R$ from the environment. Caffeine demonstrates a preference to interact with the glycerol and phosphate beads of the lipid bilayer, which is in agreement with experiments and atomistic simulations[53]. On the other hand, this behavior was not observed for adenosine. As a result, caffeine interacts with lipids prior to binding in 87% of the cases, whereas in the case of adenosine this happens in only 40% of the binding events.

A more detailed analysis of our simulations provides structural insight into the ligand-binding mechanism. In the majority of cases—all 15 for adenosine and 89 out of 142 for caffeine—the ligand firstly interacts with either ECL2 or ECL3 before reaching the binding pocket through the passage made by TM3, TM6, and TM7 (Fig. 2d, e). While adenosine directly interacts with ECL2 or ECL3 to approach $A_{2A}R$ from the water phase, caffeine is also able to establish contacts with the transmembrane helices in the initial binding phase. In such a case, caffeine interacts with ECL3 and TM6 (Fig. 2e, green solid line, 57/142 binding events) or with TM1 and TM7 entering into the cleft formed by the latter two helices (Fig. 2e, black solid line, 51/142 binding events). In 32/142 cases the ligand binds $A_{2A}R$ through ECL2 (Fig. 2e, red solid line). More details about the adenosine and caffeine binding mechanism to $A_{2A}R$ are provided in the Supplementary Discussion. An example of an adenosine binding/unbinding event to $A_{2A}R$ is provided in the Supplementary Movie 2.

As a third showcase, we studied another pharmacologically prominent member of the GPCR superfamily, the adrenergic $\beta_2$ receptor ($\beta$2AR)[54]. We successfully simulated the binding to $\beta$2AR of one natural agonist, adrenaline, and one inverse agonist, propranolol. More details about the $\beta$2AR results are given in Supplementary Fig. 5 and Supplementary Discussion.

The fourth case study is the farnesoid X receptor (FXR), a type II nuclear receptor involved in the control of bile acids, triglyceride, cholesterol, and glucose metabolism[55–57]. After activation, FXR in complex with the retinoid X receptor (RXR) binds to DNA regulating the expression of proteins involved in bile acid synthesis, triglyceride clearance, cholesterol reduction and modulating insulin sensitivity[56,57]. Therefore, in recent years FXR has become a prominent target for treatment of metabolic

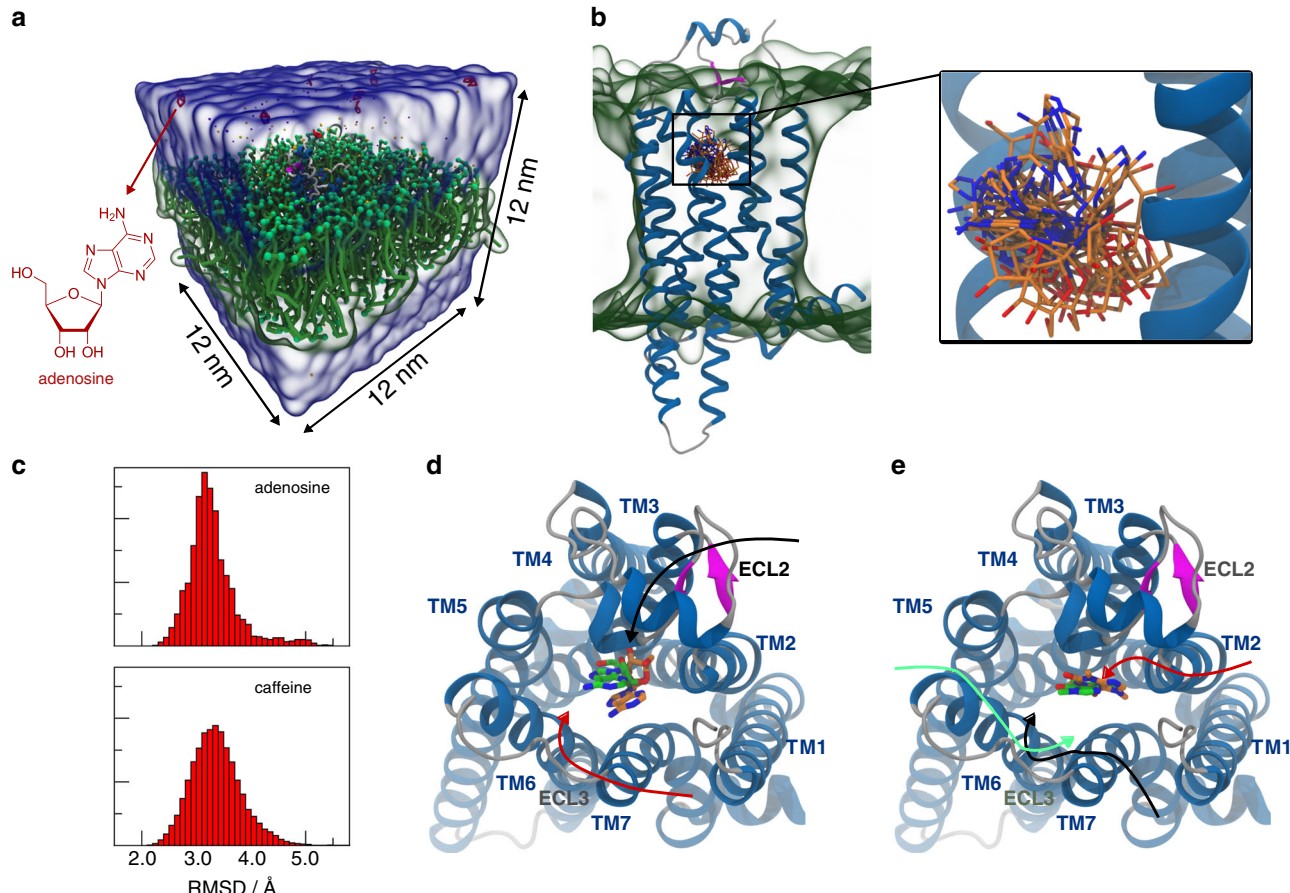

**Fig. 2 Unbiased simulations of adenosine/caffeine binding to A$_{2A}$R at the CG Martini level. a** Simulation box containing the A$_{2A}$R embedded in a POPC bilayer (green) and eight adenosine molecules (red) solvated in water (transparent blue surface). **b** Multiple binding poses of adenosine obtained from the CG model by back-mapping[50]. Only structures with an RMSD below 2.6 Å with respect to the crystallographic ligand-binding mode are shown. The structure of A$_{2A}$R is reconstructed based on the crystal structure (PDB code: 3RFM). The back-mapped ligands are shown in licorice with carbon atoms depicted in orange, nitrogen atoms in blue, and oxygens in red. **c** Histograms of the RMSD of all the binding poses of adenosine (top) and caffeine (bottom) observed during the simulations. **d, e** Comparison of the crystallographic binding mode of adenosine (**d**)/caffeine (**e**) (green) and the best binding pose obtained from CG MD simulations (orange) with the most recurring binding paths (black, red, and green arrows).

disorders, primary biliary cirrhosis, and non-alcoholic steatohepatitis syndrome[58,59]. Here, we investigate the binding of the potent agonist 6-ethyl-chenodeoxycholic acid (also known as obeticholic acid) to FXR. Obeticholic acid is a steroid acid with a specifically bent shape, containing a hydroxyl group and a charged carboxylate in the opposite extremities of the molecule (Fig. 3a). These characteristics make obeticholic acid a challenging ligand for binding studies.

In detail, we simulated 20 μs for 72 replicas of FXR in presence of 4 ligand molecules (Fig. 3a). Overall, we observe 622 binding events. The best bound pose has an RMSD of 1.2 Å and it is remarkably similar to the crystallographic one, being shifted toward helix 12 by only 1.0 Å (Fig. 3b). Figure 3c shows the RMSD values distribution of the identified binding poses computed relative to the crystal ligand-binding mode (pdb code: 1OSV)[60]. Three peaks corresponding to different conformational families are detected at 3.6 Å, 2.7 Å, and 2.0 Å, hereinafter defined as peaks 1, 2, and 3, respectively. The centroid of each family is depicted in Fig. 3d. The presence of three peaks indicates a multistep ligand-binding mechanism. We observe that obeticholic acid first approaches FXR through the loop connecting helices 5 and 6 assuming an external binding conformation (ligand at peak 1 represented as yellow sticks in Fig. 3c), and then upon the receptor rearrangement it reaches an inner binding pose (ligand at peak 2 and 3 represented as ice-blue and orange sticks,

respectively, in Fig. 3c). In all the observed binding events, obeticholic acid approaches the binding pocket contacting the loop connecting helices 5 and 6 either from the bottom, passing over helices 2 and 6 (Fig. 3b black and red arrow), or from the top through helix 1 (Fig. 3b green arrow). Such pathways resemble the so-called paths III, observed for other type II nuclear receptors[61]. In all cases, the ligand points the inner steroidal scaffold toward the binding pocket and the outer carboxylate side chain toward the solvent. This orientation is favored by the salt bridge interaction engaged by the negatively charged carboxylic group of obeticholic acid with several positively charged residues placed around the cleft, namely Arg261, Lys272, Arg328, Lys336, and Arg348 (numbering from pdb code: 1OSV), which are known to be involved in the binding of FXR agonists[62,63]. It is also interesting to note that the loop connecting helices 5 and 6 changes its conformation passing from the apo (unbound) to the holo form (ligand bound) (Fig. 3e). This leads to a reduced solvent accessible surface area (SASA) of the binding pocket computed for the apo FXR if compared with that of the holo form (Supplementary Fig. 6, blue solid line). The conformational change of the protein is further quantified by the change in the RMSD values of its backbone beads if compared with those of the crystal structure. In particular, the RMSD changes from 4.1 ± 1.0 Å in the unbound conformation to 6.9 ± 0.9 Å during ligand binding and to 5.9 ± 0.8 Å in the final bound state (blue line in

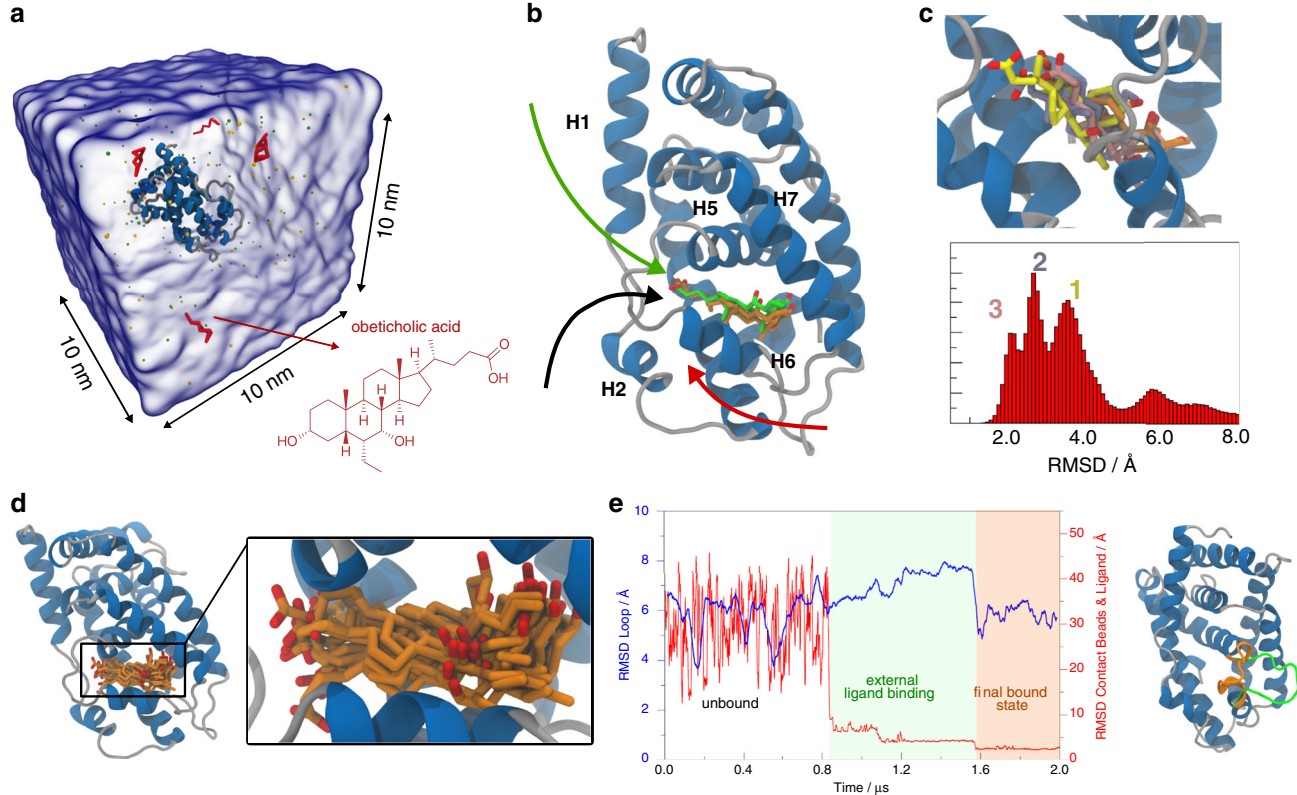

**Fig. 3 Unbiased simulations of obeticholic acid binding to FXR at the CG Martini level. a** Simulation box containing FXR and 4 obeticholic acid molecules (red) solvated in water (transparent blue surface). **b** Comparison of the crystallographic binding mode (green) and the best binding pose obtained from the CG MD simulations after back-mapping (orange). The protein is depicted in cartoon with random coils and turns colored in silver and alpha helices in blue. The molecules are shown in licorice with oxygens in red and carbon atoms in green or orange. The dominant binding pathways are shown as black, red, and green arrows. **c** Histogram of RMSD of all the binding poses observed during the simulations. The peaks at 3.6, 2.7, and 2.0 Å are denoted by the numbers 1, 2, and 3, respectively. Atomistic details of the centroids of peak 1 (yellow), 2 (ice-blue), and 3 (pink) are depicted in the protein binding pocket. **d** Multiple binding poses of obeticholic acid obtained from the CG simulations. Only structures with an RMSD below 2.3 Å with respect to the crystallographic ligand-binding mode are shown. **e** Evolution of the RMSD of the contact beads and ligand (red solid line) and the RMSD of the loop connecting helices 5 and 6 (blue solid line) during ligand binding. The plot is divided in three regions: the unbound, the external ligand binding, and the final bound state for the sake of clarity. The inset on the right shows the loop conformation in the external ligand binding (green) and final bound state (blue).

Fig. 3e). This finding together with the different SASA values of FXR in the holo and apo form, suggests an induced fit binding mechanism of obeticholic acid. Previous studies with FXR[64,65] and other type II nuclear receptors[66,67] also indicate stabilization upon ligand binding of the region around the binding site, including helices 2, 5, and 6.

Together, our results demonstrate the ability of the re-parametrized Martini CG model to reproduce experimental binding modes for a variety of receptors, even in challenging cases as GPCRs where ligand binding occurs in environments of different polarity and the nuclear receptor FXR where moderate conformational rearrangement of the binding pocket is induced by ligand binding.

**Binding to enzymes**. As a fifth showcase, we consider the human proto-oncogene tyrosine-protein kinase (c-Src) which regulates signal transduction in cells[68]. Tyrosine kinase dysfunction is linked to many diseases, and is of particular interest for cancer treatment. Regulation of tyrosine kinases is known to be governed by the presence of multiple allosteric binding sites, which have become the target of many pharmaceuticals[69]. Here, we investigate the binding of the antileukemia drug dasatinib (Fig. 4a) of which the experimental binding mode is known and its binding has been previously simulated at atomistic level[11]. Furthermore, dasatinib as a larger organic molecule compared to the previously investigated ligands, is endowed with four rings with high relative

flexibility. It represents a differently challenging system where the ligand not only needs to find the binding site but also orient itself correctly in a specific conformation. The protein was solvated in ~14,500 CG water beads, and one dasatinib molecule was added in the box, resulting in a substrate concentration of 0.96 mM (Fig. 4a). The chlorobenzene part of dasatinib binds deeply inside the highly conserved ATP pocket, while the piperazine ring remains in the outer part in contact with the solvent. Inverse binding poses, i.e. poses where the piperazine ring is buried in the active site whereas the chlorobenzene ring stays in the solvent, were rarely observed. Using a higher dasatinib concentration (i.e., five dasatinib molecules per box) lead to strong aggregation around the hinge region between the two kinase domain lobules (Supplementary Fig. 7), although binding to the ATP pocket still occurred. We found an excellent agreement in the RMSD of the bound ligand compared to the crystallographic binding pose (pdb code: 1Y57[70]), with an average RMSD of 3.0 Å (Fig. 4b), while the RMSD of the atomistic simulation was 2.0 Å[11]. 5 out of 10 independent 30 μs replicas resulted in dasatinib binding with a total of 11 individual binding events observed in the aggregate 300 μs simulation time. The binding events lasted anywhere from a few hundreds of nanoseconds to 5 μs, adding up to 8% of time spent in the bound state. Figure 4b depicts the high-density regions or dasatinib around the protein. One binding pathway was found to be coming from the region between the two protein

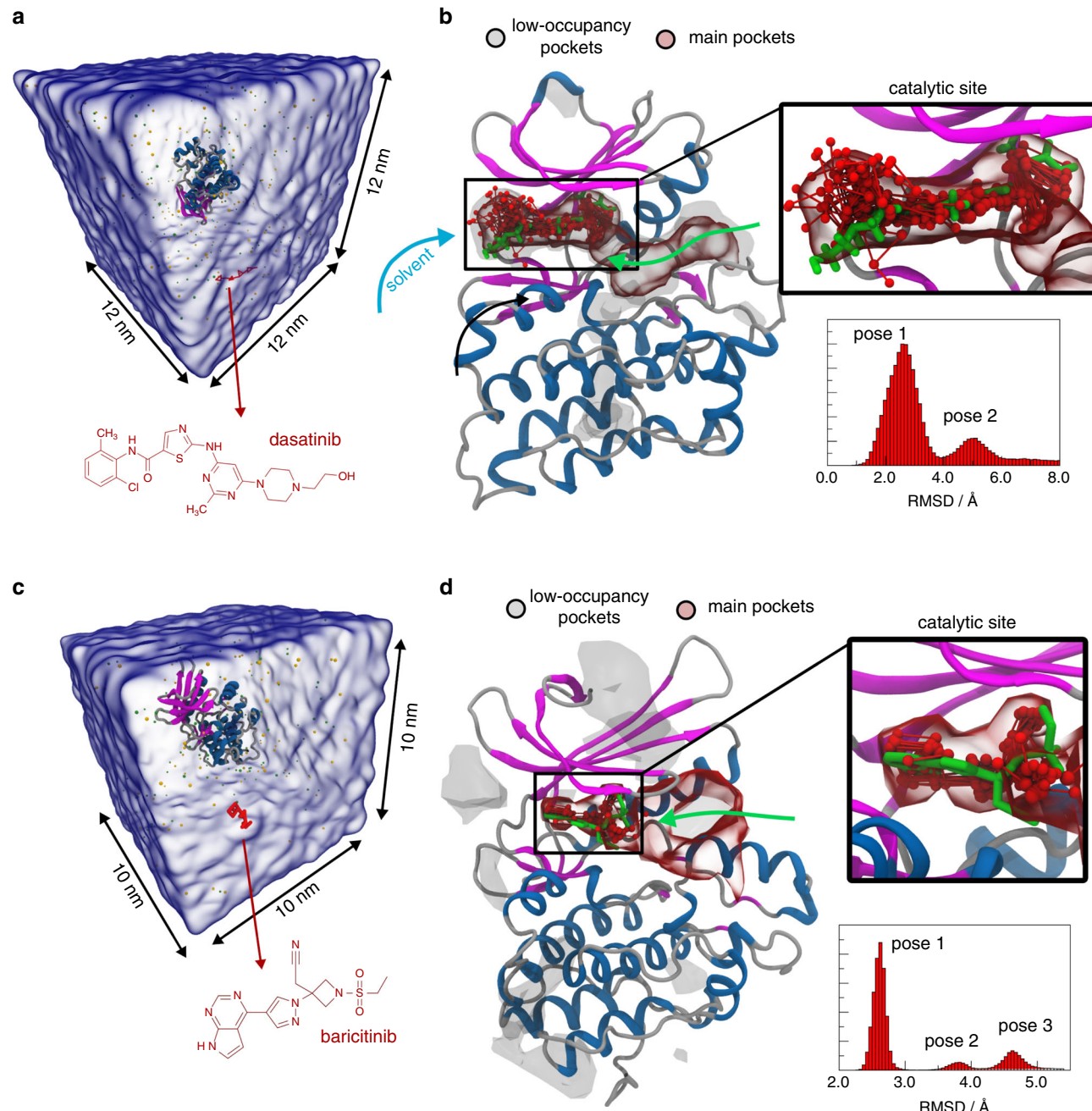

**Fig. 4 Unbiased binding simulations of dasatinib to c-Src and baricitinib to AAK1 at the CG Martini level. a** Simulation box containing the kinase domain (residues 260–533) of c-Src and one dasatinib molecule (red) solvated in water (transparent blue surface). **b** Crystal structure of c-Src (pdb code: 1Y57) with bound dasatinib (green). In addition, several CG snapshots of the bound dasatinib (red), the density in the main binding pocket (transparent red isosurface), and the density in several low-occupancy pockets (transparent gray isosurfaces) are shown. The two binding pathways observed are depicted as blue and green arrows. The histogram of the RMSD of dasatinib and the contact protein beads is depicted on the lower right. **c** Simulation box containing the AAK1 and one molecule of baricitinib (red) solvated in water (transparent blue surface). **d** Crystal structure of AAK1 (pdb code: 5L4Q) with the binding position of baricitinib taken from a homologous kinase crystal structure (pdb code: 4W9X). In addition, several CG snapshots of bound baricitinib (red), the density in the main binding pocket (transparent red isosurface), and the density in the low-occupancy pockets (transparent gray isosurface) are shown. The green arrow depicts the binding pathway. The histogram of the RMSD of baricitinib and the contact protein beads is depicted on the lower right.

lobules (Fig. 4b green arrow), in accordance with the sole dasatinib binding path identified in an atomistic simulation[11]. However, in 7 out of 11 binding events dasatinib reached the catalytic site directly from the solvent (Fig. 4b blue arrow). We estimate a $k_{on}$ of $40\,s^{-1}\,\mu M^{-1}$, which can be compared to the $k_{on}$ of $1.9\,s^{-1}\,\mu M^{-1}$ calculated from atomistic simulations[11] and $5\,s^{-1}\,\mu M^{-1}$ from experiments[71]. Considering the temporal speed-up of the Martini model due to the smoother energy landscape

(see discussion), our estimate is well in line with the expected values.

As a sixth showcase, we studied the S-selective aminotransferase of *Vibrio fluvialis* which stereoselectively catalyzes the transfer of an amino group from a donor to an acceptor ketone[72]. Using unbiased simulations we obtain good agreement of the binding pose of the substrate acetophenone. More details are provided in the Supplementary Fig. 8 and Supplementary Discussion.

The seventh and last case study presented here targets a serine/threonine kinase involved in the regulation of cellular virus endocytosis, namely the AP2-associated protein kinase 1 (AAK1)[73,74]. Targeting AAK1 has been shown to successfully prevent viral infection of cells[74] and the compound baricitinib was recently suggested as potential drug to prevent endocytosis of SARS-CoV-2 by inhibiting AAK1[34]. Baricitinib is a known inhibitor of the Janus kinases 1 and 2 JAK1/2 and is applied e.g. against rheumatoid arthritis[75]. Due to the lack of a crystal structure of AAK1 in complex with baricitinib, we used the crystal structure of AAK1 bound to the inhibitor LKB1 (pdb code: 5L4Q[76]) to generate the CG protein model, while the structure of the homolog BMP-2-inducible kinase bound to baricitinib (pdb code: 4W9X[77]) was employed as reference model for the baricitinib binding mode. Overall, the BMP-2-inducible kinase has 72% of identity with AAK1. Moreover, the catalytic site is fully conserved (100% identity), thus representing a good reference for the baricitinb binding mode to AAK1.

Figure 4c shows the simulation box containing AAK1 and one molecule of baricitinib. We performed 30 simulations of 30 μs each and observed 4 binding events to the buried catalytic site (red surface, Fig. 4d). During one of these binding events, the ligand stayed for the last 15 μs of the replica in the catalytic site. Despite the complexity of the system, we obtain an RMSD for the main binding mode of 2.6 Å with respect to the crystal pose (Fig. 4d). Two additional binding conformations are also found at farther distance from the binding site. Moreover, the ligand exhibits a tendency to bind to a more external site next to the catalytic pocket that seems to be involved in the association pathway. This pathway seems to resemble one of the pathways observed for c-Src, which may be a common feature for kinases. Several additional low-affinity pockets (gray surfaces) on the protein surface are identified, which might be relevant for the activity of baricitinib.

Together, the results presented on binding of drugs and substrates to different classes of enzymes further show the potential of our approach to predict accurate binding modes and pathways for large, flexible, and complex ligands.

## Discussion

Applications such as structure-based drug design are particularly challenging for CG modeling because of several requirements: (1) high chemical specificity of pocket-ligand interactions; (2) capability to represent all possible components of the system (as proteins, cofactors, drug candidates, solvent, lipids, etc.) in a coherent way; (3) realistic representation of conformational flexibility of each molecule in the system; (4) accurate thermodynamics and kinetics of binding. Currently, none of the CG force fields available fulfills all the requirements listed above. The examples showed in this work indicate that the current state of the Martini model seems to finally achieve most of these requirements with reasonable accuracy in relation to atomistic models. The key improvement in Martini 3[33] to enable such applications is the enhanced packing of the CG beads, achieved by re-balancing of the cross-interactions of different bead sizes[78], as well as by the re-parametrization of bonded distances based on molecular volume and shape. As a result, protein cavities are represented more realistically and ligands can fit better. Besides, the expansion of bead chemical type options in Martini 3 allows for a better coverage of the chemical space, facilitating CG modeling of rather complex small molecules such as baricitinib and dasatinib.

Given the improvements in accuracy in Martini 3, another key advantage of CG models in general is their computational performance. For instance, Martini CG based docking of biomolecular complexes can be around one order of magnitude faster compared to atomistic models, as recently demonstrated for the Haddock program[79,80]. Benchmarks tests performed with the program package Gromacs (version 2018)[81] showed that Martini based MD simulations of protein-ligand systems can be 110–350 times faster than all-atom simulations, with the performance gain increasing with growing system size (see Supplementary Discussion and Supplementary Table 2). Considering diffusion-controlled processes such as ligand binding, the smoother potential landscape of Martini can also provide 2–3 times faster association/dissociation to the proteins (Supplementary Table 3). In addition, faster protein dynamics can play an important role in induced-fit processes, as exemplified by the binding of the obeticholic acid to FXR and the dasatinib binding to c-Src kinase. The latter shows a ~8 times speed-up in association rate constant compared to atomistic simulations. Another performance advantage can be achieved by coarse-graining in the chemical space, as recently demonstrated by Menichetti et al.[82]. Because certain Martini CG moieties can represent more than one chemical fragment at the same time, virtual screening using fragment-based strategies could lead to an improvement in performance of three to four orders of magnitude. The combination of increased performance, smother potential surfaces, and coarse-graining of the chemical space can bring virtual screening protocols to speed-ups in the order of $10^5$–$10^7$ in comparison to approaches based on atomistic models.

The main limitation to achieve such a high-throughput screening pipeline is the currently limited set of available ligand parameters for Martini. The development of a curated and validated database for Martini CG ligands is therefore of paramount importance. Such ligand databases in combination with automated tools to generate CG models[83,84] (https://github.com/marrink-lab/cartographer) can expand the accessible chemical space of Martini to millions of compounds. Other aspects that can limit the accuracy of the model in certain applications, and that should be kept in mind, are: (1) the poor representation of protein flexibility by elastic network models; (2) the hydration of pockets by water molecules that cannot be represented by CG water models; (3) limited accuracy at the CG level to differentiate enantiomers or to fully represent directionality in binding poses; (4) the approximate nature of binding kinetics. Most of these issues are inherent to the process of coarse-graining, but some of these problems can be alleviated at least to some extent. For instance, protein flexibility can be greatly improved by the combination of Martini and Gō-like models[85,86]. Hydration of pockets can be modeled by the usage of smaller water beads, as it was already applied here in the $A_{2A}R$ and β2AR case (see Supplementary Methods). Polarizable Martini models[87,88] can be applied to cases for which directionally of hydrogen bonds or an improved description of the electrostatic interactions are necessary. Differentiation of enantiomers is a clear challenge for CG models, and might require two-state CG models[89] or the use of multi-resolution tools to couple Martini to all-atom models[90,91]. Further refinement of the ligand pose and the binding pocket can be achieved by back-mapping to atomistic resolution[50] as it was already used in the present study in the case of the GPCRs and FXR. Concerning kinetics, the reduction of friction from the missing atomistic degrees of freedom causes a natural speed-up of the dynamics (Supplementary Table 3). This implies that only order of magnitude estimates can be provided when using a CG model such as Martini. However, trends for ligands with the same level of resolutions can be expected to be captured well. In the case of binding, diffusional encounter between the ligand and the binding pocket entrance largely determines the binding kinetics. Because the molecular shape of the ligand, the protein, and the binding pocket entrance are represented well in Martini 3, trends

in binding kinetics are expected to be represented reasonably. Moreover, realizing that in particular unbinding rates of ligands from proteins often involve large free energy barriers, the overall accuracy of kinetic estimates for unbinding will mostly rely on a careful representation of the barrier energetics. Here, Martini is expected to perform well as the model heavily relies on reproducing free energy data and the correct kinetics rates might be retrieved from a rigorous estimate of the friction reduction[92].

In summary, we have demonstrated that the Martini 3 force field can be used to simulate protein-ligand binding in a brute force approach. We have illustrated its capability by spanning a range of systems from the well-characterized model system T4 lysozyme over pharmacologically relevant receptors such as the $A_{2A}R$, $\beta2AR$, and FXR, to a number of different enzymes, namely the aminotransferase Vf-ATA and the kinases c-Src and AAK1. In the future, the computational performance can be straightforwardly increased by optimizing the ligand concentration as well as by employing enhanced sampling techniques. Moreover, our results pave the way to an efficient computational approach to quantitatively predict binding thermodynamics and potentially capture trends in kinetics. Because no a priori knowledge of the binding pocket is required and a known pocket does not influence our approach in any way, it entails the possibility of finding new additional pockets. In this view, we could envision computational competitive binding assays. Together with ongoing developments of an automated topology builder for Martini, the presented results open the way to a high-throughput screening pipeline, potentially screening millions of drugs and protein mutants.

## Methods

**General setup for CG MD simulations.** All simulations were performed with the program package GROMACS[81] (version 2016.x or 2018.x) using the open-beta or more recent development versions of Martini 3. The beta-release of the Martini 3 model is available online at the Martini web portal[33]. Here, also the changes of the model with respect to the previous version are documented, together with the main parameter set, validation, and instruction how to use the model. More technical details about the system setups, simulation settings, and analysis are given in the Supplementary Methods and in Supplementary Tables 4–7.

**Protein CG models.** The bonded parameters of the protein models were slightly adapted from the standard Martini 2.2 settings including the recently suggested side-chain corrections[93], applied not only for β-strands but to all loops and secondary structure elements. An elastic network comparable to the one of the Martini 2.2 protein model[88] was used to maintain the secondary and ternary protein structure without exclusions of the non-bonded interactions between the backbone beads connected by the elastic network[78]. See more details about the protein models in the Supplementary Methods.

**Generic ligand parametrization.** Models for the ligands were parametrized according to the Martini (3.0) procedure: First, mappings were designed based on the following principles: (i) minimize number of CG beads used and use regular- (R-), small- (S-), and tiny- (T-) beads for 4-to-1, 3-to-1, and 2-to-1 (non-hydrogen) atoms-to-CG-site mappings; (ii) describe aromatic rings by T-beads; (iii) take into account the symmetry of the molecule. Secondly, Martini bead types were assigned based on the chemical building block they are taken to represent. Bonded interactions are then obtained based on atomistic models. Note that we used center of geometry (COG)-based mapping taking into account also the hydrogen atoms. COG-based mapping leads to better molecular (e.g., solvent accessible surface area, SASA) and bulk (e.g., mass densities) properties for the models[33,94]. Finally, models are validated using partitioning free energy data and comparison to atomistic SASA values. More details about the ligands models are given in the Supplementary Methods.

**RMSD calculation.** To calculate the RMSD between the simulated binding poses and the crystal structure, the binding pocket was aligned to the CG crystal structure. The list of residues used for the alignment of the various systems is given in the Supplementary Methods. Finally, before the RMSD calculation, the CG ligand in the crystal structure, which was obtained by transforming the bound atomistic ligand to its CG resolution, was minimized for one step to account for slight changes in the bonded parameters. Note that in case of ligands with high symmetry like benzene, all possible orientations have to be taken into account. The lowest

RMSD of all possible orientations is the correct value because all other values are too high due to flipping or rotation of the structure which does not change the chemical structure of the binding pose.

**Binding free energy calculation.** Supplementary Fig. 4 shows the radial ligand-receptor potentials of mean force (PMFs) obtained from unbiased MD simulations. One dimensional PMFs were computed based on normalized distance distributions ($p(r)$) between the center of geometry of the pocket and of the ligand, including volumetric (also called entropic) correction $2 \times \ln(r)$[95,96].

$$\mathrm{PMF}(r) = -\mathrm{RT}\ln(p(r)) + 2 \times \ln(r) \qquad (1)$$

The distance $r$ was shifted 0.25 nm to take into account the average radial volume of the ligands, which was estimated based on their average SASA values. Besides, the entire PMF was shifted to zero at 4.5 nm distance. The binding free energies (displayed in Table 1) were estimated by integrating the PMF over the distance $r$ in terms of

$$K_{\mathrm{bind}} = \int_0^{r_c} 4\pi r^2 e^{-\frac{\mathrm{PMF}(r)}{\mathrm{RT}}} \mathrm{d}r = \int_0^{r_c} 4\pi r^2 p(r)\mathrm{d}r \qquad (2)$$

$$\Delta G_{\mathrm{bind}}^0 = -\mathrm{RT}\ln(K_{\mathrm{bind}}C^0) \qquad (3)$$

with the standard concentration $C^0$ equals to (1/1.66) nm$^{-3}$. The chosen cutoff ($r_c$) was 4.0 nm, which includes binding not only in the buried pocket but also contributions from the whole protein. Based on the comparison of the number of binding and unbinding events observed in the simulations, it is possible that $\Delta G_{\mathrm{bind}}^0$ obtained for systems such as ethylbenzene and n-propylbenzene are slightly underestimated, because around 35% of the simulations did not show any unbinding by the end of the trajectories.

**Reporting summary.** Further information on research design is available in the Nature Research Reporting Summary linked to this article.

## Data availability
Data supporting the findings of this paper are available from the corresponding authors upon reasonable request.

## Code availability
Force-field parameters and procedures (e.g. tutorials) are publicly available at http://cgmartini.nl.

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

## Acknowledgements

We thank Dr. Ignacio Faustino and Dr. Jonathan Barnoud for their effort in the initial development of Martini 3 protein and adenosine models. S.T. acknowledges the support from the European Commission via a Marie Skłodowska-Curie Actions individual fellowship (MicroMod-PSII, grant agreement 748895). J.C.R.P. thanks the CONACYT program for a doctoral fellowship. P.C.T.S., S.T., J.C.R.P., R.A., and S.J.M. acknowledge the National Computing Facilities Foundation (NCF) of The Netherlands Organization for Scientific Research (NWO) for providing computing time. V.L. acknowledges the support from the Swiss National Science Foundation (Project No. 200021_163281), the Italian MIUR-PRIN 2017 (2017FJZZRC), and the COST action CA15135 (Multitarget paradigm for innovative ligand identification in the drug discovery process MuTaLig). V.L. also acknowledges the support by a grant from the Swiss National Supercomputing Centre (CSCS) under project ID s712 and the Partnership for Advanced Computing in Europe (PRACE) for awarding access to project ID 2016153685.

## Author contributions

P.C.T.S. designed the research with suggestions from S.J.M., S.T., and V.L.; P.C.T.S., P.C., J.C.R.P., R.A., and S.R. performed the simulations; all authors analyzed the data. The paper was written by S.T., S.J.M., P.C.T.S., P.C., and V.L. with contributions of all other authors. All authors have given approval to the final version of the paper.

## Competing interests

The authors declare no competing interests.
