## [Peer Review File · Nature Communications]

Peer Review File: Reviewers' comments first round:

Reviewer #1 (Remarks to the Author):

Review of NCOMMS-19-28960

This is an interesting study of the use of Martini (version 3) coarse-grained molecular dynamics simulations to explore millisecond timescale sampling of ligand/drug binding to 3 protein targets. The results are very promising with respect to reproducing available experimental data, and suggest that this approach may have considerable value in the future in novel approaches to rapid MD based screening of ligand/target protein interactions. The authors are experts in the development of coarse-grained simulations for proteins and membranes, and the simulations and analysis are performed to a high standard. I think perhaps the phrase 'mark a milestone on the way' is a bit hyped (and some reference to the comparable use of atomistic simulations for fragment screening e.g. Arcon et al. (2017) *J. Chem. Inf. Model.* should be included) but this is an important step forward and merits publication following revision.

The approach is illustrated via its application to three test systems: T4 lysozyme, a well characterized model system, S-selective aminotransferase (ATA), and the A2a receptor as an example of a pharmacologically important model target. Whilst I can understand why just 3 examples were studied in a proof-of-principle study, I remain unclear as to why ATA was selected as one of these rather than selecting a better-characterized pharmacologically important target protein.

For T4 lysozyme, 4 structurally similar ligands (benzene, phenol, indole, and n-propylbenzene) – were explored, and gave excellent agreement with experiments, both structurally and in terms of free energies. This is very encouraging.

For the second example - the S-selective aminotransferase (ATA) – it is not clear that there are suitable experimental data against which to evaluate the simulation results.

The third example is a membrane inserted protein, the adenosine A2A receptor, and the crystallographic binding poses of adenosine and caffeine are reproduced in the simulations.

So, overall this is very promising, and in addition to predicting binding sites/poses and free energies of interaction, details of multiple interaction pathways are predicted. However, to be certain of the future value of the method it may be necessary to probe a wider range of chemical space (in terms of ligands) and of protein target space, alongside validation against a wider range of experimental data.

One more 'technical' aspect which requires some amplification is that of the degree of flexibility of the protein targets. In the SI it is mentioned (page S6) that, as is standard for Martini protein simulations, an elastic network was applied to the protein. It is also noted (page S9) that a shorter cutoff was used for the GPCR study. It would be interesting to know to what extent the results of the screening simulations were dependent on e.g. the cutoff for the elastic network. In the atomistic simulation literature (e.g. Lexa and Carlson, 2010, *J. Amer. Chem. Soc.*) it has been shown that correct representation of protein flexibility is important for hot-spot mapping by MD simulations.

Reviewer #2 (Remarks to the Author):

The authors are the well-known developers for MARTINI coarse-grained models for various biomolecules. The coarse-grained models have been utilized in many biological simulations, such as biological membranes, and membrane proteins. In this work, the authors used the updated MARTINI model for protein-ligand binding simulations. Considering the reduced computational cost, this approach has great possibility in in-silico drug discovery and should be useful for many computational scientists.

Overall, I have good impressions about the work, however, some details are now well written. I would like to point out my concerns in the work one by one.

1) In the work, the authors described only the positive aspects in the protein-ligand binding. However, coarse-graining of the models brings several limitations. So, the authors should describe the limitation of the current approach in protein-ligand binding.

For instance,
the accuracy of the binding mode, i.e. ligand-sidechain interaction accuracy
kinetics of binding processes
protein conformational change upon ligand binding, in particular, for larger ligands

2) The authors indicated that this approach is successful to search the ligand binding sites without knowing any experimental knowledge. However, the previous simulation studies suggests that in long MD trajectories, not only the canonical binding sites but also other weaker binding sites are predicted. For instance, JACS 133, 9181-9183 (2011).

I would like to know how to discern the canonical and other binding sites in the approach.

3) It is helpful to show MARTINI model for each ligand and the most successful binding poses (with sidechain of the target protein). This information is necessary for readers to understand 'their' successful ligand binding prediction.

4) Related to the third point, if the atomistic binding poses cannot be simulated by this coarse-grained model, is it possible to refine the binding pose by using atomistic force fields ? Or this is out of scope in this work ?

I believe that this is not straightforward, since the binding pocket is narrow and ligand may not be able to do rotational and translational motions in the narrow binding pocket. I would like to hear a practical strategy for converting the coarse-grained model to atomistic model, if the prediction is not perfectly fine.

5) In Figure 1, T4 lysozyme structure is shown. It is helpful to add notation of helices (F, G, H, etc) for defining the binding pathways. The notation of helices in T4 lysozyme is established in previous works and should be common in the work. Also, conformational fluctuation of T4 lysozyme (and other two proteins) should be shown in supporting information. In T4 lysozyme, the excited conformation was found by NMR spectroscopy (Nature 2011, 477, 111-114). In the simulation, was the conformation sampled or not ?

6) In Figure 2, I believe that Fig. 1A should be smaller and the comparison between the binding pose prediction and experimental structure should be add. Fig. 2B is not clear for the purpose.

7) I have interest on the conformational changes of the receptor, with or without ligand binding. How can the ligand go into the deep inside of membrane is not a simple question. I believe that more extensive analysis and data are necessary for the conformational changes, ligand-binding pathways, and competition with water. Again, I would like to know not only the canonical binding sites but also other weak binding sites, if they found.

8) Terada et al. previously published protein-ligand binding simulation with the MARTINI models. The authors should cite the paper and discuss the difference between the current and their works. J Comput Chem. 2014 Sep 30;35(25):1835-45. doi: 10.1002/jcc.23693. Epub 2014 Jul 21

9) It is helpful if the authors discuss which update in MARTINI v.3 is the most important for

improving the prediction accuracy.

Reviewer #3 (Remarks to the Author):

Presented are calculations using Course Grained (CG) Simulations of the binding of a simple ligands to three different receptors. The method is simply to use MD to allow the ligands to diffuse to and interact with their binding pockets. While this may be the first example of using CG for such calculations, the results are as expected and no insights into ligand-protein interactions are presented. Furthermore, indicating that this method will be of utility for drug design is certainly not supported by the present results.

The calculations use ligands with simple ring system with only one or two (if any) rotatable bonds amongst all the ligands as "drugs." These are certainly not drug-like molecules and, given that they are relatively hydrophobic in nature it can be expected with high probability that they will find the hydrophobic sites on the protein to which they are known to bind. More importantly, there is absolutely no evidence that the course grained method has the structural resolution to differentiate the subtle changes in drug-protein interactions that lead to changes in drug affinity and specificity, including changes in off rates, associated with modification of the ligand structures. Accordingly, the present results show nothing more than the course grained method can identify binding pockets on proteins for very simple molecules using brute force simulations; the work certainly does not show that the method will be of any utility to drug design. This would require validation against significant numbers of ligands for a number of different proteins for which experimental affinities and binding orientations are known. There are many such examples in the literature that the authors can follow when designing such a study.

Following are some issues they authors may considering expanding on in the study.

On line 185, the authors discuss on rates for ligands. This is largely a diffusion controlled event, with the rate of diffusion in the present study being enhanced by the use of small ligands. Can the authors make a quantitative estimate of the on rates from the simulations and discuss those rate with respect to the diffusion controlled limit. It would be interesting to understand how different the on rate is with T4 lysozyme given the need for the binding pocket to open to become allow for ligand binding.

While the authors have only studied a few ligands, it would be interesting if they made a quantitative comparison of the CG results and experimental binding data for the compounds. Are they getting equilibrium values close to the experimental estimates? Are the relative affinities for the few ligands that are studied in agreement with experiment?

Detailed response to the reviewers' comments:

Reviewer #1:

This is an interesting study of the use of Martini (version 3) coarse-grained molecular dynamics simulations to explore millisecond timescale sampling of ligand/drug binding to 3 protein targets. The results are very promising with respect to reproducing available experimental data, and suggest that this approach may have considerable value in the future in novel approaches to rapid MD based screening of ligand/target protein interactions. The authors are experts in the development of coarse-grained simulations for proteins and membranes, and the simulations and analysis are performed to a high standard. I think perhaps the phrase 'mark a milestone on the way' is a bit hyped (and some reference to the comparable use of atomistic simulations for fragment screening e.g. Arcon et al. (2017) J. Chem. Inf. Model. should be included) but this is an important step forward and merits publication following revision.

Authors' reply: We thank the reviewer for the positive comments. Despite including more protein examples such as a nuclear receptor and two kinases as well as significantly more complex drug-like ligands such as propranolol, obeticholic acid, dasatinib, and baricitinib, we reformulated the phrase "The presented results mark a milestone on the way for potential high-throughput screening [...]". It now reads: "The presented results open the way to high-throughput screening [...]".

In addition, we included the suggested reference in the introduction on page 3 (reference number 41).

The approach is illustrated via its application to three test systems: T4 lysozyme, a well characterized model system, S-selective aminotransferase (ATA), and the A2a receptor as an example of a pharmacologically important model target. Whilst I can understand why just 3 examples were studied in a proof-of-principle study, I remain unclear as to why ATA was selected as one of these rather than selecting a better-characterized pharmacologically important target protein.

For T4 lysozyme, 4 structurally similar ligands (benzene, phenol, indole, and n-propylbenzene) – were explored, and gave excellent agreement with experiments, both structurally and in terms of free energies. This is very encouraging.

For the second example - the S-selective aminotransferase (ATA) – it is not clear that there are suitable experimental data against which to evaluate the simulation results.

Authors' reply: We agree that the availability of experimental reference data for the S-selective aminotransferase is limited and added two additional examples of enzymes with available

experimental data (c-Src kinase, AAK1) as well as two pharmacologically well-characterized receptors (adrenergic beta2 GPCR and farnesoid X nuclear receptor).

The example of an ATA addresses another aspect within the pool of studied proteins: It is an enzyme of high interest from a synthetic chemists point of view because it stereoselectively catalyzes the transfer of an amino group. This makes it an interesting target for computational enzyme design, for which we consider our new approach very useful as well.

The third example is a membrane inserted protein, the adenosine A2A receptor, and the crystallographic binding poses of adenosine and caffeine are reproduced in the simulations.

So, overall this is very promising, and in addition to predicting binding sites/poses and free energies of interaction, details of multiple interaction pathways are predicted. However, to be certain of the future value of the method it may be necessary to probe a wider range of chemical space (in terms of ligands) and of protein target space, alongside validation against a wider range of experimental data.

***Authors' reply:** We thank the reviewer again for the positive judgement of our presented approach. To address the potential future value, we included several additional protein-ligand systems and substantially expanded the diversity of chemical groups being present in the ligands. The examples discussed in the revised manuscript entail the most prominent benchmark system for protein-ligand binding studies, namely L99A mutant of T4 lysozyme, examples from the three very popular pharmaceutical target families, G-protein coupled receptors, nuclear receptors, and kinases, as well as a synthetically interesting enzyme. In the case of T4 lysozyme, our data demonstrates not only the possibility to discriminate between binders and non-binders, but we also achieve a quantitative agreement of the binding free energy of nine protein-ligand systems. Moreover, the pool of pharmaceutical targets contains examples from proteins which make up more than 50% of the current small-molecule drug targets (Santos, et al., Nat. Rev. Drug Discov. **16**, 19 (2017)). Specifically, regarding the chemical space of the ligands, we have acknowledged the reviewer's suggestion performing additional investigations on differently sized ligands, from fragments (e.g. benzene with 78 Da) as in the case of T4 lysozyme, to middle size drug-like ligands (e.g. propranolol with 259 Da) and larger drug molecules (e.g. dasatinib with 488 Da). The new results are now reported and discussed in the revised manuscript.*

One more 'technical' aspect which requires some amplification is that of the degree of flexibility of the protein targets. In the SI it is mentioned (page S6) that, as is standard for Martini protein simulations, an elastic network was applied to the protein. It is also noted (page S9) that a shorter cutoff was used for the GPCR study. It would be interesting to know to what extent the results of the screening simulations were dependent on e.g. the cutoff for the elastic network. In the atomistic simulation literature (e.g. Lexa and Carlson, 2010, J. Amer. Chem. Soc.) it has been shown that correct representation of protein flexibility is important for hot-spot mapping by MD simulations.

Authors' reply: We fully agree with the reviewer that testing the effect of the elastic network settings on the binding properties of the ligands is a very interesting aspect. However, a systematic testing of this aspect goes beyond the scope of the current work.

The decision to use a shorter cutoff for the elastic network for the GPCR study was based on the fact that in the case of GPCRs the protein flexibility is an important aspect for the ligand binding. In particular, the flexibility of the extracellular loops ECL1, ECL2, or ECL3 is relevant to the binding kinetics as proven in different experimental and computational studies [Manglik, A. et al., Nature 485, 321–326 (2012), Dror, R. O. et al., Proc. Natl. Acad. Sci. U. S. A. 108, 13118–13123 (2011)]. Receptor activation (i.e. binding to G proteins) requires even more flexibility and thus is not investigated in our study.

In a long-term perspective, the use of Martini 3 together with Gō-like models seems to be promising in order to realistically represent protein flexibility. As an example, a recent study on Cu,Zn superoxide dismutase [Souza, P. C. T. et al., J. Phys. Chem. Lett. 10, 7740–7744 (2019)] showed that it is possible to unravel allosteric pathways over a distance of ~2 nm using the open-beta version of Martini 3 combined with a Gō-like model.

Reviewer #2:

The authors are the well-known developers for MARTINI coarse-grained models for various biomolecules. The coarse-grained models have been utilized in many biological simulations, such as biological membranes, and membrane proteins. In this work, the authors used the updated MARTINI model for protein-ligand binding simulations. Considering the reduced computational cost, this approach has great possibility in in-silico drug discovery and should be useful for many computational scientists.

Overall, I have good impressions about the work, however, some details are now well written. I would like to point out my concerns in the work one by one.

Authors' reply: We thank the reviewer for the positive comments.

1) In the work, the authors described only the positive aspects in the protein-ligand binding. However, coarse-graining of the models brings several limitations. So, the authors should describe the limitation of the current approach in protein-ligand binding.

For instance, the accuracy of the binding mode, i.e. ligand-sidechain interaction accuracy kinetics of binding processes protein conformational change upon ligand binding, in particular, for larger ligands

Authors' reply: We thank the reviewer for pointing out the necessity of a section discussing the advantages and limitations of the presented approach. We added the section "Advantages and limitations" before the Conclusions on page 22 of the manuscript. In brief, we discuss the following advantages:

- packing of the CG beads leading to a better representation of the protein side chains and ligands.
- expansion of chemical bead types which allows to model the complex chemical groups being present e.g. in dasatinib and baricitinib
- computational performance (see also Supplementary Methods, Supplementary Tables S2 and S3) resulting in a speedup of 10^5 - 10^7 compared to atomistic models.

As well as the following limitations:

- currently limited set of available ligand models.
- limited protein flexibility (addressable by using Gō-like model).
- hydration of protein pockets (addressable by using tiny water beads).
- limited accuracy of CG models for enantiomers.
- approximate nature of binding kinetics (however, the trends of binding as well as unbinding are expected to be captured reasonably).

For more details, please see page 22ff of the manuscript.

2) The authors indicated that this approach is successful to search the ligand binding sites without knowing any experimental knowledge. However, the previous simulation studies suggests that in long MD trajectories, not only the canonical binding sites but also other weaker binding sites are predicted. For instance, JACS 133, 9181-9183 (2011).

I would like to know how to discern the canonical and other binding sites in the approach.

Authors' reply: We fully agree with the reviewer's comment that by using unbiased sampling not only the main binding pockets will be discovered, but also weaker binding sites can be detected. We would like to point out that the fact that free sampling has no bias for any kind of pockets is one of the major advantages of the work presented here. It allows to detect all protein-ligand interaction sites which are potentially of importance for disclosing the binding mechanism of a drug molecule. Moreover, the obtained computational speedup in sampling of 10^2 - 10^3 compared to atomistic models allows us to obtain converged ensembles for the protein-ligand interactions. This results in realistic relative populations of the observed binding sites.

From a biophysical point of view, there is no a priori difference between a canonical binding site compared to a non-canonical one. Moreover, how can we be sure that experimental procedures are able to correctly predict the ligand binding site competent for the drug pharmacodynamics? It might be the case that the binding site or even binding mode detected in a crystal structure is equally important for the pharmacodynamics as another which is suppressed. Suppression might occur due to crystal packing because protein crystallization might impact the selection of one binding mode out of a pool of available poses which are similarly populated. This was for instance

reported in the case of the cyclooxygenase-2 enzyme in which two alternative binding modes were found by simulations and fluorescence quenching experiments, although only one was detected by crystallography (Limongelli et al. *Proc Natl Acad Sci U S A.* **107**, 12, 5411-5416 (2010); Lanzo et al. *Biochemistry.* **39**, 20, 6228-6234 (2000)).

In the examples selected in our present study, experiments have shown that the main pockets detected in the crystal structure and have the highest affinity are at the same time the key pockets responsible for affecting the protein's function. This entails that our approach has to be able to detect them which indeed is the case. However, it does not necessarily exclude binding to other low-affinity binding sites as e.g. shown for the kinases c-Src and AAK1.

3) It is helpful to show MARTINI model for each ligand and the most successful binding poses (with sidechain of the target protein). This information is necessary for readers to understand 'their' successful ligand binding prediction.

Authors' reply: *We added Supplementary Figure S1 showing the structure of each of the 15 studied ligands together with their CG models.*

The RMSD histograms shown in Fig.s 1B, 2C, 3C, 4B/D as well as in the Supplementary Fig.s S5 and S8 provide quantitative information about the quality of the sampled binding poses. They show that the obtained accuracy of the sampled binding pose is on the order of the resolution in the crystal structure. There is not a single structure which is the final result of the present studies. Rather the RMSD distribution in the binding site is a result which gives information about the poses adapted while the ligand is in the pocket. Nevertheless, selected examples of best ligand binding poses are given for the GPCRs and the nuclear receptor.

4) Related to the third point, if the atomistic binding poses cannot be simulated by this coarse-grained model, is it possible to refine the binding pose by using atomistic force fields ? Or this is out of scope in this work ?

I believe that this is not straightforward, since the binding pocket is narrow and ligand may not be able to do rotational and translational motions in the narrow binding pocket. I would like to hear a practical strategy for converting the coarse-grained model to atomistic model, if the prediction is not perfectly fine.

Authors' reply: *We agree with the reviewer that subsequent atomistic refinement of the binding poses obtained using the CG Martini force field can provide valuable additional information. Of course, this refinement potentially entails a number of challenges like e.g. narrow pockets, difficulty of ligand rotations and position adjustments. However, similar challenges occur when refining docking poses with atomistic MD and thus the applied strategies can be employed here in a similar manner. Potentially, smart solutions can be developed in the future which, e.g., allow slow conformational changes while backmapping from the CG to the atomistic model, or utilize*

an ensemble of CG structures instead of a single optimal one. Yet, another option would be to use Hamiltonian exchange or other multi-resolution tools to couple Martini to all-atom models. For the initial introduction of atomistic details into configuration obtained with the CG Martini force field an efficient tool exists - backward.py (Wassenaar, et al., J. Chem. Theory. Comput. 10, 676 (2014)) - which is publicly available at the Martini web portal www.cgmartini.nl. In order to demonstrate this powerful approach, atomistic refinement has already been applied in the present study in the cases of the A_{2A} and the β 2 receptors as well as the nuclear receptor FXR.

5) In Figure 1, T4 lysozyme structure is shown. It is helpful to add notation of helices (F, G, H, etc) for defining the binding pathways. The notation of helices in T4 lysozyme is established in previous works and should be common in the work. Also, conformational fluctuation of T4 lysozyme (and other two proteins) should be shown in supporting information. In T4 lysozyme, the excited conformation was found by NMR spectroscopy (Nature 2011, 477, 111-114). In the simulation, was the conformation sampled or not ?

Authors' reply: *We thank the reviewer for this suggestion and added the common helix labels in Fig. 1D of the manuscript.*

The excited state conformation is an interesting aspect of the benchmark system T4 lysozyme and it would be nice to investigate the presence of this conformation in our CG model in more detail. In the experimental study mentioned by the reviewer, it is shown that only 3% of the population exists in this excited conformation. This is usually beyond the targeted accuracy in any CG modelling. Moreover, according to the referred paper, the ligand binding occurs only via the ground state.

Thus, this excited state conformation is of minor importance for the purpose of the current investigation and its investigation is beyond the scope of the current manuscript.

We finally point out that protein conformational changes are observed in the case of the nuclear receptor FXR in which the ligand binding induces a rearrangement of the region around the binding site. This is analyzed and discussed in the main text of the revised manuscript.

6) In Figure 2, I believe that Fig. 1A should be smaller and the comparison between the binding pose prediction and experimental structure should be add. Fig. 2B is not clear for the purpose.

Authors' reply: *Unfortunately, to the best of our knowledge, there is no experimental structure available for the binding pose of the substrate in the enzyme active site of ATA. Therefore, it is only possible to provide a comparison with the structure optimized at the QM/MM level of theory. In order to overcome the lack of an experimental reference structure, we added additional examples of enzymes (c-Src kinase and AAK1). Moreover, we shifted the focus of the enzyme section to the two kinase examples where we highlight the comparison between experimental and*

simulated binding pose. Most of the discussion of ATA is moved to the Supplementary Discussion now, because we consider the example as being of importance for the computational enzyme design community.

7) I have interest on the conformational changes of the receptor, with or without ligand binding. How can the ligand go into the deep inside of membrane is not a simple question. I believe that more extensive analysis and data are necessary for the conformational changes, ligand-binding pathways, and competition with water. Again, I would like to know not only the canonical binding sites but also other weak binding sites, if they found.

***Authors' reply:** These are all very exciting points raised by the reviewer. However, a detailed discussion about all these topics is beyond the scope of the paper. Our main focus is to show that the Martini 3 CG model can find the binding pockets and pathways with reasonable accuracy. Nevertheless, in this new version of the paper, we tried to reinforce these aspects pointed by the reviewer in some of the examples presented along the text. Conformational changes are an important part of the FXR-obeticholic acid showcase. Examples of the interplay between binding pockets, water and bilayers are presented for the A2A and $\beta 2$ receptors. The existence of low occupancy (weak) pockets is shown in both kinases examples. We hope this glimpse of such important features is enough to indicate that Martini 3 may be capable to be used in the study of these different aspects involved in the protein-ligand binding process.*

8) Terada et al. previously published protein-ligand binding simulation with the MARTINI models. The authors should cite the paper and discuss the difference between the current and their works.

***Authors' reply:** We are aware of three studies published in peer-reviewed journals using Martini models for protein-ligand binding simulations: two studies from the group of Prof. Tohru Terada (Negami et al., *J. Comput. Chem.*, 2014; and Negami et al., *Chem. Phys. Lett.*, 2020) and the study of Jiang and Zhang, published in *J. Chem. Inf. Model.*, 2019. It is important to note that most of the examples explored in these papers are exceptional cases, where the authors used some controversial choices in the models which can create artifacts in the results observed. For instance, to avoid packing problems (probably in the pockets), Negami et al 2014 added additional bonds between the side chains using force constants of only 30 kJ/mol. We recently published a study (Alessandri et, *J. Chem. Theory Comput.*, 2019) showing that bond force constants lower than 400 kJ/mol can end up collapsing the beads involved in these harmonic potentials. So, the pockets in the study of Negami et al. (2014) are probably only available because of this issue.*

The study of Jiang and Zhang do not seem to force the Martini 2 model to work as the previous example mentioned. However, in this case, the association of sphingosine-1-phosphate was induced by placing the ligand at the entrance of the pocket. So, no free sampling was performed to observe the binding event. More importantly, this lysosphingolipid was placed with its C16 tail pointing to the water phase, in a very unrealistic orientation for this binding process. The most

probable choice would be this lipid be associated to the GPCR from the bilayer. A similar situation is found in our study for the adenosine and caffeine binding to the A_{2A} receptor. At variance with the work of Jiang and Zhang, we did not apply any bias to the entry of the ligand to the receptor, performing free sampling starting with the ligand in the fully solvated state. Our CG simulations are able to let the ligand find its preferential route to the binding site engaging interactions with interfaces of different polarity known to be involved in the receptor/ligand binding. Despite the fact that the discussed examples apply problematic settings for the protein model or unrealistic starting conditions for the binding process, which both can result in serious artifacts, we now cite the works considering their efforts in the field.

9) It is helpful if the authors discuss which update in MARTINI v.3 is the most important for improving the prediction accuracy.

Authors' reply: We agree with the reviewer. The key improvements in Martini 3 to enable its potential usage in structure-based drug design are now described in the new "Advantages and limitations" section (as described in point 1, reviewer #2). Moreover, together with the release of the open-beta version of the Martini 3 force field, the changes in comparison to the previous version are explained.

Reviewer #3:

Presented are calculations using Course Grained (CG) Simulations of the binding of a simple ligands to three different receptors. The method is simply to use MD to allow the ligands to diffuse to and interact with their binding pockets. While this may be the first example of using CG for such calculations, the results are as expected and no insights into ligand-protein interactions are presented. Furthermore, indicating that this method will be of utility for drug design is certainly not supported by the present results.

The calculations use ligands with simple ring system with only one or two (if any) rotatable bonds amongst all the ligands as "drugs." These are certainly not drug-like molecules and, given that they are relatively hydrophobic in nature it can be expected with high probability that they will find the hydrophobic sites on the protein to which they are known to bind. More importantly, there is absolutely no evidence that the course grained method has the structural resolution to differentiate the subtle changes in drug-protein interactions that lead to changes in drug affinity and specificity, including changes in off rates, associated with modification of the ligand structures. Accordingly, the present results show nothing more than the course grained method can identify binding pockets on proteins for very simple molecules using brute force simulations; the work certainly does not show that the method will be of any utility to drug design. This would require validation against significant numbers of ligands for a number of different proteins for which experimental affinities

and binding orientations are known. There are many such examples in the literature that the authors can follow when designing such a study.

***Authors' reply:** We agree with the reviewer in the sense that our study is not (yet) a proof that the coarse-grain methodology presented can be applied to drug-design. We consider the current work as an important step toward this direction. Admittedly, we do not generate new insights and, as the reviewer rightfully claims, most results are 'as expected'. However, in our view (and shared by the other reviewers), this is exactly what makes our method valuable: with only a fraction of the computational cost, we reproduce results obtained with more detailed atomistic models. Even for some of the simple ligands considered, this is not as trivial as the reviewer claims, as it not only involves prediction of the correct binding pocket and binding pose, but also reproducing known binding pathways as well as binding free energies.*

To alleviate the point raised by the reviewer that the set of ligands studied was somewhat limited, we extended our study with more examples, including ligands that are polar, differently sized, more drug-like, and contain rotatable bonds. Please see also the response to reviewer 1 for a more detailed account on which systems were added.

Finally, we are fully aware of the potential limitations of our method, including the limited structural resolution that can be achieved. We admit that this aspect was not properly discussed in the original manuscript, and have added a discussion on this topic in the revised manuscript.

Following are some issues the authors may be considering expanding on in the study.

On line 185, the authors discuss on rates for ligands. This is largely a diffusion controlled event, with the rate of diffusion in the present study being enhanced by the use of small ligands. Can the authors make a quantitative estimate of the on rates from the simulations and discuss those rates with respect to the diffusion controlled limit. It would be interesting to understand how different the on rate is with T4 lysozyme given the need for the binding pocket to open to become allow for ligand binding.

***Authors' reply:** Assessment of quantitative rates with CG models is difficult. Due to the removal of atomistic degrees of freedom, the potential energy landscape is smoothed resulting in an overall speedup of the dynamics. Although this is advantageous with respect to the sampling efficiency, this prevents accurate quantification of ligand binding rates. We now discussed this limitation in the revised manuscript. At a more qualitative level, however, rates can be calculated, and provide order of magnitude estimates that appear to be in the right ballpark as we show in case of the c-Src kinase example. A more systematic evaluation of binding and unbinding rates, and comparison of trends between different ligands, would be very interesting but we consider them out of the scope of the current manuscript.*

While the authors have only studied a few ligands, it would be interesting if they made a quantitative comparison of the CG results and experimental binding data for the compounds. Are they getting equilibrium values close to the experimental estimates? Are the relative affinities for the few ligands that are studied in agreement with experiment?

Authors' reply: This is a good suggestion. We now performed an extensive analysis of the binding free energies of the nine systems involving T4 lysozyme, and found very good agreement with known experimental data. The results are presented in Table 1 of the revised manuscript.

The power of our method is also given by the fact that it can be combined with other binding free energy techniques, like funnel-metadynamics or free-energy perturbation, to reach a quantitative calculation of the absolute and relative ligand binding free energy to locally defined binding sites. This advance is currently under investigation in our labs and will be the argument of an upcoming publication.

Editor

Before I can make a decision, I have a couple of questions that need to be clarified. I do see many MD sim paper but I am by heart a trained biochemist/biophysicist. So please oversee if some of my questions / comments are not valid points.

1) you write “ we show that the recently re-parameterized Martini model (38) can perform this task... “ But I am confused since Ref 38 is unpublished. Wouldn't it be extremely important to have access as a reviewer but also as a reader to the model to know what was re-parameterized?

Authors' reply: The open-beta version of Martini 3 which is used in most of the cases presented here, is available online already (www.cgmartini.nl). Together with the parameters, explanations are available discussing the changes of the new model in comparison to Martini 2, its validation, as well as instructions on how to use the new model. Thus, the readers have access to everything they need in order to apply our presented approach. In addition, we now discuss the key changes/improvements of the new force field with respect to the previous version focusing on the application in protein-ligand binding in a dedicated section named “Advantages and limitations” of the revised manuscript and reported all the details about the setup of the Martini models in the Supplementary Discussion.

2) Which leads me to my next question. The usability/applicability for other researchers. Would researchers be able to use the re- parameterized model for their own research questions based on the methods section in your manuscript? Will there be a GUI or an upgrade/tutorial at <http://cgmartini.nl/index.php>?

***Authors' reply:** Together with the revised manuscript, we submitted two nice tutorials for users to set up simulations similar to the ones reported in the presented work and an introduction on how to parametrize a new ligand molecule. These tutorials are now included for review only, but we will of course include them on the Martini web portal as soon as this manuscript is published. We will also provide all the necessary input files required to repeat the simulations reported in the current manuscript.*

3) Would there be a way to compare how resource intense your previous atomistic simulations where compared to the new approach? I think a direct comparison of required run time or another measure would be a strong addition.

***Authors' reply:** Thank you for the suggestion, this is a very good idea. As said, in the revised manuscript we included a section discussing the advantages and limitations of our approach, in which we also address this point. Moreover, we added more details in the Supplementary Discussion as well as in Supplementary Tables S2 and S3.*

REVIEWERS' COMMENTS second round:

Reviewer #1 (Remarks to the Author):

Review of NCOMMS-19-28960A Revised Paper

The authors have responded to nearly all of my comments satisfactorily in the rebuttal and revised manuscript.

In particular, they have expanded the range of their study which helps to convince the reader of the potential of the methods they have developed.

I do think that their response to the question of elastic network settings is a bit weak – some degree of testing of the effect of varying the elastic network cut-off would be easy to do, and indeed they presumably explored this at a preliminary level to decide upon the shorter cut-off for GPCRs. However, I do not think this should be a sticking point. They refer to possible advances by using Martini 3 + Go-like models, both in the rebuttal and the revised ms. It will be of interest to see how this performs for these systems in future studies.

In summary, I think the revised manuscript is much stronger and I would recommend acceptance.

Reviewer #2 (Remarks to the Author):

This is my second review about the manuscript on the use of MARTINI coarse-grained models for protein-ligand binding. I found that the revised manuscript has been improved and the authors answered questions and comments appropriately.

Prediction of protein-ligand or protein-drug interactions is one of the central issues in computational chemistry. Atomistic approaches can be more reliable if sufficient sampling is achieved. But, protein-ligand binding process is slow and atomistic MD simulations may not be able to predict the binding process with reasonable computational resources.

The authors proposed in the paper the use of MARTINI coarse-grained models and showed successful binding results. Upon the request of the previous review, the authors expanded the discussion in the manuscript, even the limitation of current models. I believe that this is a significant progress in the field and would contribute to in-silico drug discovery also. I suggest the publication of the manuscript in the current form.

Reviewer #3 (Remarks to the Author):

The authors have addressed my concerns.